# Synthesis, Characterization, and DFT Calculations of a New Sulfamethoxazole Schiff Base and Its Metal Complexes

**DOI:** 10.3390/ma16145160

**Published:** 2023-07-21

**Authors:** Jibril I. Al-Hawarin, Abdel-Aziz Abu-Yamin, Abd Al-Aziz A. Abu-Saleh, Ibrahim A. M. Saraireh, Mansour H. Almatarneh, Mahmood Hasan, Omar M. Atrooz, Y. Al-Douri

**Affiliations:** 1Department of Chemistry, Al-Hussein Bin Talal University, Ma’an 71111, Jordan; 0306013@ahu.edu.jo (J.I.A.-H.); dribrahim9997@gmail.com (I.A.M.S.); 2Department of Chemistry, Memorial University, St. John’s, NL A1B 3X7, Canada; aabusaleh@mun.ca (A.A.-A.A.A.-S.); m.almatarneh@ju.edu.jo (M.H.A.); 3Department of Chemistry, University of Jordan, Amman 11942, Jordan; 4Hepi Company (Home of Experience) for Paints and Inks, Cairo 61710, Egypt; hassan_sci@yahoo.com; 5Department of Biological Sciences, Mutah University, Mutah 617102, Jordan; 6Nanotechnology and Catalysis Research Center (NANOCAT), University of Malaya, Kuala Lumpur 50603, Malaysia; 7Department of Mechanical Engineering, Faculty of Engineering, Piri Reis University, Eflatun Sk. No: 8, Istanbul 34940, Tuzla, Turkey; 8Department of Applied Physics and Astronomy, College of Sciences, University of Sharjah, Sharjah P.O. Box 27272, United Arab Emirates

**Keywords:** Schiff base, characterization, 3-(2-furyl)acrolein, sulfamethoxazole, DFT, lanthanide, zinc complexes, antioxidant, anti-inflammatory, anti-hemolytic

## Abstract

A new Schiff base, 4-((1E,2E)-3-(furan-2-yl)allylidene)amino)-N-(5-methylisoxazol-3-yl) benzene-sulfonamide (L), was synthesized by thermal condensation of 3-(2-furyl)acrolein and sulfamethoxazole (SMX), and the furan Schiff base (L) was converted to a phenol Schiff base (L’) according to the Diels–Alder [4 + 2] cycloaddition reaction and studied experimentally. The structural and spectroscopic properties of the Schiff base were also corroborated by utilizing density functional theory (DFT) calculations. Furthermore, a series of lanthanide and transition metal complexes of the Schiff base were synthesized from the nitrate salts of Gd, Sm, Nd, and Zn (L**1**, L**2**, L**3**, and L**4**), respectively. Various spectroscopic studies confirmed the chemical structures of the Schiff-base ligand and its complexes. Based on the spectral studies, a nine-coordinated geometry was assigned to the lanthanide complexes and a six-coordinated geometry to the zinc complex. The elemental analysis data confirmed the suggested structure of the metal complexes, and the TGA studies confirmed the presence of one coordinated water molecule in the lanthanide complexes and one crystalline water molecule in the zinc complex; in addition, the conductivity showed the neutral nature of the complexes. Therefore, it is suggested that the ligand acts as a bidentate through coordinates to each metal atom by the isoxazole nitrogen and oxygen atoms of the sulfur dioxide moiety of the SMX based on FTIR studies. The ligand and its complexes were tested for their anti-inflammatory, anti-hemolytic, and antioxidant activities by various colorimetric methods. These complexes were found to exhibit potential effects of the selected biological activities.

## 1. Introduction

Schiff bases are well-known ligands that are made by the condensation of primary amines with carbonyl groups (aldehyde or ketone), which is called an imine or azomethine group with the general formula R_2_C = N-R’, where R and R’ are alkyl, aryl, and/or heterocyclic groups. Hugo Schiff reported them in 1864 in the nineteenth century [1,2]. Schiff bases may be mono-, bi-, or tridentate ligands capable of generating highly stable complexes with transition metals in addition to lanthanides. Chelating agents through polydentate donor sites such as the SMX Schiff base acting as a tridentate ligand resulted in stable complexes [3]. In comparison, 2-amino-6-ethoxybenzo-thiazole may be exposed as a monodentate or bidentate ligand. SMX forms several aromatic Schiff bases with drug carrier applications and the photostability of PVC applications [4]. Schiff bases and metal complexes may play a crucial role in exhibiting various biological and pharmacological properties [5]. Others have reported many biological activities, including antifungal [6,7,8], antibacterial [9,10], antimalarial [11], antipyretic [12,13,14], antiproliferative [15], antiviral [6,14], anti-inflammatory [16], and antitubercular agent (Lippard) properties [17]. These activities are due to the azomethine group. Schiff bases contain an azomethine group obtained through the condensation of primary amines with carbonyl compounds [18]. These biological activities of Schiff bases are due to their ability to form complex compounds with metals in the active site of numerous enzymes involved in metabolism [19]. The potential pharmacological properties of Schiff bases may be attributed to the formation of complex compounds with metal ions in the active site of many enzymes involved in metabolism [20,21]. In addition, the nitrogen atom of the imine (-C=N-) group is suggested to form a hydrogen bond with the active sites of cell constituents, which interferes with normal cell processes [22]. Moreover, the inhibition of aminoacyl-tRNA synthesis pathways by these Schiff bases may be another reason for their biological activities [23,24]. Antioxidants are substances that can prevent, or slow down, cell damage caused by free radicals—unstable molecules produced by the body in response to environmental and other stresses—sometimes called “free radical scavengers” [25]. Sources of antioxidants can be natural or synthetic [26]. Oxidation originally meant adding oxygen to a compound and losing an electron. When losing an electron from an atom, a free radical is created, and free radicals can damage tissues and DNA [27]. The antioxidant can accept that electron or hydrogen radical and become stable. Inflammation is a localized physical condition in which part of the body becomes hot, red, swollen, and painful, especially in response to an injury or infection [28]. The main classes of anti-inflammatory agents are glucocorticoids and non-steroidal anti-inflammatory drugs (NSAIDs) [25]. NSAIDs are the most widely used anti-inflammatories. NSAIDs inhibit the production of prostaglandins, a group of compounds that contribute to inflammatory response and are responsible for signs such as fever and pain. NSAIDs are very effective, and unlike other anti-inflammatories, they have no immunosuppressive effect [29]. Hemolysis is caused by the breakdown of red blood cells because of membrane lipid bilayer rupture. Such hemolysis seems related to extract quantity and effectiveness. The main limitation or challenges to the therapeutic effect of some Schiff bases is that they are reported to inhibit or interact with cytochrome P450 isoenzymes, therefore, leading to toxic or adverse effects due to the lower clearance and may be an accumulation of the drug or its metabolites [30], in addition to the hepatotoxicity and toxicological endpoints [20,31].

In addition to their wide range of industrial applications [32], Schiff bases are present in various natural and non-natural compounds. Moreover, all thiazole Schiff bases have been studied widely due to their great ability to form complexes with several metals, either transition metal or lanthanides, with great physical properties such as luminescence [33,34], magnetism [35], and crystallography structures [36]. In this work, the synthesis and characterization of a new Schiff base are investigated, as compared with experimental results with theoretical data using density functional theory (DFT) to support the formation of the compound. In addition, the biological activities of the new Schiff base are characterized by its antioxidant, anti-inflammatory, and anti-hemolytic properties.

## 2. Experimental Methods

### 2.1. Materials and Methods

All chemicals were obtained from the Aldrich (Saint Louis, MO, USA), TEDIA (Fairfield, OH, USA), and Merck (Rahway, NJ, USA) companies. The melting point of the synthesized Schiff base was determined by employing an electrothermal melting point SMP 10 apparatus. The FTIR spectrum was detected using an FTIR Bruker-ATR (Leipzig, Germany). A Bruker Avance III −500 MHz was used to record the ^1^H and ^13^C NMR spectra in DMSO solvent. The mass spectra were recorded on a Brucker apex-IV. The UV–visible spectrum of the Schiff base–DMSO solution was recorded in the range from 200 to 900 nm using a SPECORD PLUS by Analytik Jena AG. GCMS ISQ models from Thermo Scientific were used for direct inlet mass spectrometry, and the FLASH 2000 CHNS/O analyzer was used to determine the elemental contents of the synthesized compounds.

### 2.2. Synthesis Procedure of the Schiff Base, C_17_H_15_N_3_O_4_S

As illustrated in Figure 1, 0.489 g and 4 mmol of 3-(2-furyl)acrolein were dissolved in 10 mL of ethanol; 2–3 drops of concentrated glacial acetic acid was added [37] as a catalyst into this solution, and the solution was stirred for 10 min at room temperature. An equimolar amount of SMX (1.01 g, 4 mmol) was added to this solution. The color changed to bright brown, and the solution was refluxed for 3 h. After that, the furan Schiff base was converted to a phenol Schiff base by the intramolecular Diels–Alder reaction of furans because the furan as the dienic component is less numerous due to a reluctance of the aromatic ring to undergo [4 + 2] cycloaddition [38,39] as shown in Figure 1. Afterward, the solvent was evaporated in vacuum to about 50%, and the pale brown solution was then left standing to precipitate through slow solvent evaporation at room temperature. After three days, a brown powder was obtained. The crystals were washed three times with EtOH (5 mL each) and dried in a vacuum at 35 °C for 3 h to give (E)-4-((3-hydroxybenzylidene)amino)-N-(5-methylisoxazol-3-yl)benzenesulfonamide as a yellowish-brown Schiff base compound.

Yield, 1.2 g (3.36 mmol, 84%), Mp: 131–132 °C. Anal. calc. for C_17_H_15_N_3_O_4_S (357.38 g/mol): expected (found); C, 57.13 (55.40); H, 4.23 (4.30); N, 11.76 (11.34); S, 8.97 (7.86). HR-ESI-MS (positive ion mode) *m*/*z*: calc. for C_17_H_16_N_3_O_4_S (M + H)^+^ 358.08560, found 358.08679, FTIR (cm^−1^): 3462, 3373_w_, 3206_m_, 3155_w_, 3101_w_, 3082_w_, 2972_w_, 1661_m_, 1611_w_, 1593_s_, 1502_s_, 1463_s_, 1374–1389_s_, 1326_s_, 1156_vs_, 925_s_, and 882_w_. ^1^H NMR (500 MHz, DMSO-d6) δ: 9.54, 10.89, 4.29, 8.10–7.26, 7.53–6.62, 8.25, 6.53, 6.05, 3.29 ppm and ^13^C NMR (126 MHz, DMSO-d6) δ: 193.96, 153.7, 170.3, 150.7, 95.7, 156, 121.9–132.9, 146.1, 113.0, and 113.8.

### 2.3. General Procedure for the Preparation of L**1**–**4**

The Schiff base ligand, 4-((1E,2E)-3-(furan-2-yl)allylidene)amino)-N-(5-methylisoxazol-3-yl) benzene-sulfonamide (L) in 20 mL of ethanol (1.0 g, 2.80 mmol), was added in a 1:2 molar ratio dropwise over 10 min at ambient temperature into a 10 mL solution of [Ln(NO_3_)_3_.6H_2_O] (metal 1–3) (1.40 mmol) and into 10 mL of [Zn(NO_3_)_2_.4H_2_O] (metal 4) (1.40 mmol). Following refluxing of the reaction mixture for 3 h, the solid obtained was washed with cold ethanol (3 × 10 mL) and diethyl ether (3 × 10 mL). Recrystallization of the solid from hot ethanol yielded the compounds L**1**–**4**, as shown in Figure 2.

#### 2.3.1. [Gd(L)_2_(NO_3_)_3_(H_2_O)] (L**1**)

A pale brown powder was obtained in a yield of 78.61% (1.2 g, 1.10 mmol). Mp: > 300 °C. Anal. calc. for C_35_H_35_N_9_O_18_S_2_Gd (1091.08 g/mol): C: 37.95, H: 3.00, N: 11.72, S: 5.96. Found, C: 38.79, H: 3.06, N: 12.04, S: 5.89. FTIR (ATR, ν cm^−1^): 3462 (OH) 3354 (N–H), 1640 (C=N)azomethine, 1588 (C=N)isoxazole ring, 1372 (C–N); NO_3_ bands: 1498 (*ν_4_*), 1319 (*ν_1_*), 1030 (*ν_2_*), 882 (*ν_6_*), 738 (*ν_3_*), 695 (*ν_5_*).

#### 2.3.2. [Sm(L)_2_(NO_3_)_3_(H_2_O)] (L**2**)

A pale brown powder was obtained in a yield of 72.52% (1.10 g, 1.01 mmol). Mp: 240–244 °C. Anal. calc. for C_35_H_35_N_9_O_18_S_2_Sm (1084.19 g/mol): C: 38.20, H: 3.02, N: 11.79, S: 6.00. Found, C: 38.91, H: 2.73, N: 11.89, S: 5.87. FTIR (ATR, ν cm^−1^): 3462 (OH) 3342 (N–H), 1642 (C=N)azomethine, 1588 (C=N)isoxazole ring, 1374 (C–N); NO_3_ bands: 1489 (*ν_4_*), 1306 (*ν_1_*), 1022 (*ν_2_*), 879 (*ν_6_*), 730 (*ν_3_*), 695 (*ν_5_*).

#### 2.3.3. [Nd(L)_2_(NO_3_)_3_(H_2_O)] (L**3**)

A brown powder was obtained in a yield of 79.56% (1.2 g, 1.10 mmol). Mp: >245–248 °C. Anal. calc. for C_35_H_35_N_9_O_18_S_2_Nd (1078.07 g/mol): C: 38.42, H: 3.03, N: 11.86, S: 6.03. Found, C: 38.79, H: 3.98, N: 11.97, S: 6.03. FTIR (ATR, ν cm^−1^): 3462 (OH), 3348 (N–H), 1642 (C=N)azomethine, 1584 (C=N)isoxazole ring, 1370 (C–N); NO_3_ bands: 1463 (*ν_4_*), 1318 (*ν_1_*), 1028 (*ν_2_*), 883 (*ν_6_*), 737 (*ν_3_*), 695 (*ν_5_*).

#### 2.3.4. [Zn(L)_2_(NO_3_)_2_(H_2_O)]H_2_O (L**4**)

A pale yellow powder was obtained in a yield of 76.27% (1.0 g, 1.07 mmol). Mp: >300 °C. Anal. calc. for C_35_H_35_N_8_O_15_S_2_Zn (937.21 g/mol): C: 44.28, H: 3.50, N: 12.15, S: 6.95. Found, C: 42.03, H: 3.40, N: 13.28, S: 6.51. FTIR (ATR, ν cm^−1^): 3462 (OH) 3354 (N–H), 1658 (C=N)azomethine, 1582 (C=N)isoxazole ring, 1366 (C–N).

## 3. Computational Methods

All DFT calculations of the structural and spectroscopic properties of Schiff base were conducted using the Gaussian 16 package [40]. Thirty-one different conformers of the compound were generated using the Confab tool of the Open Babel package with the default cutoff (RMSD = 0.5 Å and energy = 50 kcal/mol) [41].

These conformers were optimized using the M06-2X/6-31G(d) level of theory [26,27]. The most stable conformer with the lowest-energy conformation was considered for IR, NMR, and UV–Vis. The M06-2X functional was reported to perform very well for the geometries, kinetics, non-covalent interactions, and thermochemistry of the main group elements and yielded reliable results compared to experimental data and high-level ab initio calculations [42,43,44,45,46,47].

For IR calculations, the double hybrid functional B2PLYP [48] and the def2-TZVP basis set were used. The scaling factor of 0.9995 was applied for the calculated harmonic frequencies. It has been reported that the B2PLYP/def2-TZVP level of theory performed very well for the calculation of harmonic frequencies [49]. The UV–Vis absorption energies of the Schiff bases were calculated using the time-dependent-density functional theory (TD-DFT method (TD-M06-2X/6-311G(2d,p) level of theory).

For NMR calculations, the LC-TPSSTPSS/cc-pVTZ [50,51] level of theory was used. A previous benchmark study reported that the long-range corrected functional, specifically the LC-TPSSTPSS, and the Dunning correlation-consistent polarized triple-ζ basis set (cc-pVTZ) are recommended for NMR calculations [52]. The continuous set of gauge transformations (CSGT) method outperformed the Gauge-independent atomic orbital (GIAO) method [52]. Therefore, the CSGT method was used to calculate the chemical shifts. The tetramethylsilane (TMS) compound was selected as a reference for the NMR spectra. It should be mentioned that the IR calculations were performed in the gas phase, while the DMSO PCM solvation model was used for NMR and UV–Vis calculations to mimic the real environment of the solvated Schiff base compound.

## 4. Biological Activities

### 4.1. Determining Total Phenol Content

The sample’s total phenol content was calculated with the help of the modified Folin–Ciocalteu test. Firstly, a mixture of 0.1 mL of the extract’s supernatant and 0.5 mL of Folin–Ciocalteu’s reagent was prepared and incubated at room temperature for 5 min. Then, a 2.5 mL aqueous solution of sodium carbonate was added to the extract, and the absorbance was measured with a UV–Vis spectrophotometer at 765 nm. Various concentrations of gallic acid resulted in the creation of a calibration curve. The overall phenol content was assessed using the calibration curve for gallic acid, as described by Chlopicka et al. [53].

### 4.2. DPPH Radical-Scavenging Assay

To measure the plant extract’s antioxidant activity, a reaction mixture was prepared with the combination of 50 μL of the plant extract with 1.0 mL of DPPH (2,2-diphenyl-1-picryl-hydrazyl-hydrate) solution. This mixture was incubated at room temperature for 30 min, and then, the absorbance was measured against a blank sample of methanol solution at 517 nm. Gallic acid acted as a positive control, and the inhibition percentage was determined based on the equation below from Bal et al. [54].

Inhibition activity (%) = (Ac-As) × 100/Ac, where Ac is the control’s absorbance, and As is the sample’s absorbance.

The half-maximal inhibitory concentration (IC_50_) is defined as the concentration at which DPPH radicals are scavenged at about 50% and is calculated from the concentration–response curves. To determine the IC_50_ value, the AAA Bioquest IC_50_ calculator was utilized.

### 4.3. Determining Anti-Inflammatory Activity

To determine the extract’s anti-inflammatory activity, a colorimetric method described by Naz et al. [55] was used. The solution was prepared by adding 0.05 mL of each extract to 0.45 mL of bovine serum albumin (BSA), while 0.05 mL of the sample in 0.45 mL of distilled water acted as a product control. In the first step, 1.0 mg/mL of diclofenac standard stock solution was incubated at 37 °C for 20 min; in the second step, it was incubated at 70 °C for 10 min. Before checking the absorbance at 660 nm, 2.5 mL of phosphate buffer was added to the solution.

The anti-inflammatory activity was measured according to the following equation:Anti-Inflammatory activity (%) = 100 − (AT − (AP/AC)) × 100,
where AT is test solution; AP is product control; AC is test control. To determine the IC_50_ value, the AAA Bioquest IC_50_ calculator was utilized.

### 4.4. Determining Anti-Hemolytic Activity by H_2_O_2_ Method

To assess the anti-hemolytic activity, different concentrations of each extract (0.25 mL) were combined with 1.0 mL of RBCs and 1.25 mL saline and incubated for 10 min at 37 °C before adding 0.23 mL of H_2_O_2_. The incubation was extended for further 2 h at room temperature, followed by centrifugation for 10 min at 3000 rpm. After centrifugation, the absorbance was measured at 540 nm. The anti-hemolytic activity was calculated using the formula below provided by Naz et al. [55].

Anti-hemolytic activity (%) = (Ap–(As-Ac/Ap)) × 100, where Ap is the positive control’s absorbance, Ac is the negative control’s absorbance, and As is the sample’s absorbance. To determine the IC_50_ value, the AAA Bioquest IC_50_ calculator was utilized.

## 5. Results and Discussion

### 5.1. Geometric Optimization

The optimized geometry of the most stable ground-state conformer for the Schiff base is shown in Figure 1. For more details, the Cartesian coordinates (in Å) of the optimized structure are available in the Appendix A in Appendix A.

### 5.2. FTIR Studies

#### 5.2.1. FTIR of the Schiff Base (L)

The spectral data were in good agreement with the newly prepared Schiff base. The measured and calculated IR spectra of the Schiff base are depicted in Figure 2. Also, the measured and computed wavenumbers of the Schiff base are listed in Table 1. The calculated IR spectrum for the Schiff base agreed with the measured IR spectrum, as shown in Figure 2. The IR spectrum of the Schiff base exhibited the following characteristic absorption: the observed band at 3462 cm^−1^ for OH of phenol and 3376 cm^−1^ for the free NH_2_ group of SMX [56] disappeared concurrently with the new observed band at 1661 cm^−1^ due to the C=N imino stretching vibrations, which confirmed the consistency of the Schiff base. All other SMX bands [56] were not much more affected by the Schiff base formation. The band shown at 3206 cm^−1^ is assigned to the sulfonamide NH stretching vibration [57]. In addition, methyl antisymmetric and symmetric stretching was observed at 3155 cm^−1^ and 3101 cm^−1^, respectively. The band observed at 2973 cm^−1^ refers to the ether group in furan, and the absorption band at 1611 cm^−1^ refers to the CN isoxazole ring. The strong absorption band at 1593 cm^−1^ refers to phenyl ring C=C stretching [58], and isoxazole ring vibration corresponds to 1374 and 1389 cm^−1^. The strong absorption band at 1326 cm^−1^ refers to the antisymmetric vibration of SO_2,_ while the symmetric vibration appears sharp at 1156 cm^−1^. Finally, the absorption bands at 1593 and 1502 cm^−1^ are due to the aromatic furan moiety.

#### 5.2.2. FTIR of the Schiff Base (L**1**–L**4**)

Comparing the FTIR spectra of the complexes L**1**, L**2**, L**3**, and L**4** and the free ligand as shown in Appendix A (in the Appendix A), the weak broad band at 3206 cm^−1^ in the ligands is assigned to the (NH) group. This band remains almost at the same position in the complexes and hence suggests that the (NH) group nitrogen does not take part in coordination. However, the band related to the isoxazole ring stretching vibrations in free ligand at 1389 cm^−1^ undergo a shift to 1372, 1374, 1370, and 1366 cm^−1^ in the spectra of the L**1**, L**2**, L**3**, and L**4** complexes, respectively, indicating that the isoxazole moiety is participating in coordination with metal ions [59]. The shift of two sulfonamide vibrations (the symmetric as well as the asymmetric one [60]) toward a lower shift number in the spectra of the complexes as compared to the spectra of corresponding ligands (Table 2) further supports (M-O) bonding [3]. The new bands observed at 430–572 cm^−1^ are tentatively assigned to ν (M-N) and (M-O) (metal–ligand) [61,62]. The metal complexes show a broad band in the region 3467, 3387, and 3397 cm^−1^ and a new band at 825, 833, and 825 cm^−1^ that is assigned to the (O-H) stretching vibration and out-of-plane bending of water molecules coordinated to the complexes L**1**, L**2**, and L**3**, respectively [63]. The appearance of a band at 1661 cm^−1^ in the ligands is assigned to ν(C=N) azomethine; this band remains almost at the same position in the complexes and hence suggests that the azomethine nitrogen is not taking part in coordination [4].

The complexes contained two distinct molecules: L**1**, L**2**, and L**3** with three bidentate nitrates and L4 with two bidentate nitrates. Absorption bands of the coordinated nitrates were observed at 1463–1498 (ν as) and 879–883 (ν as) cm^−1^. The free nitrate ν3 (E′) appears at 1306–1319 cm^−1^ for the L**1**, L**2**, and L**3** complexes, respectively. In addition, the separation of the two highest-frequency bands (ν 4–ν 1) is approximately 160 cm^−1^, and accordingly, the coordinated NO_3_^−^ is bidentate [59].

### 5.3. NMR Analysis of the Schiff Base

The ^1^H and ^13^C NMR main signal chemical shifts of the Schiff base were recorded in DMSO as a solvent; Appendix A (in the Appendix A). The chemical shifts are reported in parts per million (ppm). The NMR spectra of the Schiff base were also calculated using the B3LYP/6-311+G(2d,p) level of theory. Several studies show that DFT calculations provide valuable insights into the ^1^H and ^13^C NMR chemical shifts, aiding in the interpretation and prediction of spectroscopic data [64,65,66].

The chemical shift at δ = 9.54 ppm suggested the attribution of the proton of the CH = N imine group. Remarkably, there was a broad downfield chemical shift at 10.89 for (OH) and a peak at 4.29 ppm for NH. The chemical shift of the aromatic proton of the benzene ring was observed within the 8.10–7.26 ppm region of the spectrum. In comparison, the chemical shift of the furan ring protons was observed within the 7.53–6.62 ppm region of the spectrum. Remarkably, the downfield chemical shift (δ = 8.25 ppm) corresponds to the proton of alkene in the ortho position of the furan ring. It has a higher electron affinity than the other proton of the alkene in the same bond, which appears at δ = 6.53 ppm. The singlet chemical shift at 6.05 ppm refers to the isoxazole from SMX moiety; finally, the methyl group from SMX was seen at 3.29 ppm.

The ^13^C NMR of the Schiff bases in DMSO was used as a complementary technique for confirming the compound formation. The ^13^C NMR (DMSO-d6, δ, ppm): the spectrum of the compound was characterized by a sharp peak at 193.96 (carbon of phenol), 153.7 (–N=CH–), 170.3 (–C–O from isoxazole moiety), 150.7 (–C=N from isoxazole moiety), 95.7 (CH= from isoxazole moiety), and 156 (refer to C-N=). The other benzene carbons were at 121.9–132.9 ppm; the chemical shift at 146.1 was for C-O from the furan ring. The other two carbons of the furan ring were at 113.0 and 113.8, and the methyl groups appeared at 12.5 ppm. Table 3 lists the observed and calculated chemical shifts. Figure 3 shows the atom labeling of the Schiff base that is mentioned in Table 3. Mainly, the calculated NMR chemical shifts overestimated the observed results. This difference could be due to the fact that the implicit solvation model did not mimic the real explicit environment of the DMSO solvent. In other words, the PCM solvation model lacks the description of solute–solvent hydrogen bonds explicitly between the donor (the Schiff base) and the acceptor (the DMSO), which results in a significant difference between the calculated and observed chemical shifts. It should be also noted that there was a significant contrast between the calculated and the observed ^1^H-NMR of the sulfonamide proton (labeled as H12, see Table 3). This disagreement between the experimental and theoretical results for the sulfonamide proton was also reported by Dayan and coworkers [51].

### 5.4. UV–Vis Analysis of Schiff Base

The experimental and theoretical UV–Vis spectra of the Schiff base were determined. We performed TD-DFT using the M06-2X/6-311G(2d,p) level to determine the excited states of the Schiff base. We summarized the lowest-energy excited states of the compound in Appendix A (see the Appendix A). The measured and calculated UV–Vis spectra of the Schiff base are shown in Figure 4. The results show that the calculated λ_max_ of 352 nm is in excellent agreement with the experimental value of 348 nm, differing by no more than 4 nm. The first excited state is the strongest peak, which has the most significant value of oscillator strength (see the Appendix A). This state is defined as a HOMO to LUMO transition. The contribution from other molecular orbitals is small. Figure 5 describes the HOMO and LUMO of the Schiff base. It is evident that the electron density moves from the furan ring (donor side of the molecule) to the sulfonyl group (acceptor side). We also plotted the electrostatic potential of the compound and calculated the different densities between the excited state minus the ground state (see Figure 6). In this figure, the blue region implies where the difference density is positive (i.e., the electron density of the excited state is larger than it is in the ground state). In contrast, the red region indicates where the difference density is negative. Therefore, the iso-surface gives in-depth information about the redistribution of the electron density and the nature of electronic transitions occurring from the furan ring (the red part) to the opposite end of the compound (the blue part) as a transition from the ground state to the first excited state.

### 5.5. Mass Spectroscopic Study

The HR-ESI-MS (positive ion mode) spectrum of the Schiff base (L + H)^+^ with the molecular formula C_17_H_16_N_3_O_4_S showed its molecular ion *m*/*z* at 358.08679 compared with a calculated value (M + H)^+^ of 357.07833 This result is supported further by elemental analysis and TGA measurements. The observed molecular ion peak (m+) at *m*/*z* 358.08679 is due to the corresponding Schiff base, while the other peaks refer to its monoisotopic isomers as shown in Appendix A (in the Appendix A).

### 5.6. Thermal Analyses

Thermogravimetric analysis (TGA) was carried out to determine the thermal stability of the complexes L**1**, L**2**, L**3**, and L**4** under a flow of nitrogen. The TGA curves are shown in Figure 7.

The TGA data for all the La(III) complexes showed two decomposition steps within the temperature range of 25–900 °C. Similar TGA screens were noticed for all the lanthanide complexes, with very few differences in the weight loss for each decomposition event, while the zinc behaved differently.

The first small decomposition occurred in the temperature range from 25 to 200 °C, which mainly refers to the loss of coordinated water molecules from the inner coordination sphere of the L**1**–L**3** complex (for complexes <2.5 wt%) and for crystalline water from zinc complex as its loss increases a little bit after 95 °C. The second large decomposition step within the temperature range 235–390 °C accounts for the removal of NO_3_^−^ ions in the form of N_2_O_5_, as reported in the literature for related complexes [67]. The third stage corresponds to the decomposition of the complex moiety with increasing temperature, leaving about 45% as a residue after the decomposition steps and the organic loss of the complexes is due to the formation of La_2_O_3_ (Ln = Gd, Sm, and Nd) and, with respect to the zinc complex, it is about 34% for ZnO and ash [67].

### 5.7. Biological Activities

Chemical extracts have biologically active molecules with excellent pharmacological properties. Therefore, the present study focused on the characterization of the pharmacological properties like anti-hemolytic, antioxidant, and anti-inflammatory.

The total phenolic content (TPC) was estimated using Folin–Ciocalteu’s method, and gallic acid was used as a standard. The results of the TPC are illustrated in Table 4. The findings of the current study revealed that the chemical extracts of these chemicals had a significant quantity of phenolic contents compared to the gallic acid used as the reference. Our explanation is that this may be due to the presence of phenolic rings in the structure of these chemicals.

The antioxidant activity of chemical extracts was found to possess different levels of potential efficiency by using the DPPH radical scavenging method. It was found that the chemical L4 possessed the highest percentage of activity. In addition, the IC_50_ regression value was between 0.5015 and 0.7372 µg/mL.

The antioxidant results of the current study showed that chemical extracts had a moderate activity of DPPH radical scavenging, and the IC_50_ value was efficient at low quantities. The higher antioxidant activity is related to the type of phenolic rings present in the chemical structure [68].

The chemical extracts were found to be highly active and possessed about 99.0% anti-inflammatory property, except for L, which had a lesser activity level of 83.94%, while the IC_50_ regression result for the chemical extract was between 0.1738 and 0.748 µg/mL. Furthermore, the anti-hemolytic activity experiment was performed to determine whether these chemicals could neutralize the targeting of red blood cells by H_2_O_2_ and thus prevent oxidative damage to the erythrocyte membrane. The results showed a clear and different inhibition of hemolysis by the chemicals, where L**1** exhibited weak activity of 6.56 ± 0.001%, while L, L**4**, L**2**, and L**3** exhibited a moderate activity of anti-hemolysis (%) 45.75 ± 0.002, 27.23 ± 0.003, 40.7 ± 0.001, and 37.01 ± 0.001 respectively. Furthermore, the IC_50_ regression results ranged from 0.4768 to 0.8081 µg/mL. The current study showed unexpected results that these chemicals had high anti-inflammatory activity and can be used as a treatment for inflammation. In addition, the anti-hemolytic activity findings showed that the chemicals had moderate inhibition of hemolysis of erythrocytes.

The tested compounds exhibited a membrane stabilization effect to prevent red blood cell (RBC) membrane lysis by inhibiting hypotonicity. The RBC membrane is equivalent to a lysosomal membrane, and the selected compounds may also stabilize lysosomal membranes [69]. The stabilization of lysosomal membranes plays a significant role in limiting the inflammatory response by preventing the lysosomal constituents’ release in the activated phagocytes such as proteases and other bactericidal enzymes, which are causes of inflammation [70]. Furthermore, the prevention of protein denaturation, particularly, albumin denaturation, and inhibitory effects on cyclooxygenase may contribute to anti-inflammatory activity [71].

In general, Schiff bases or metal complexes exert their effects by inhibiting enzymes, enhancing lipophilicity, interacting with biomolecules, arresting the cell cycle, and altering cell membrane functions [72]. For example, many metal complexes of the quinolone group of some antibiotics are reported to possess enhanced activity compared to antibiotics alone [73] and suggest potential avenues for further improvement [74,75,76,77,78,79,80].

## 6. Conclusions

A new Schiff base derived from 3-(2-furyl)acrolein and SMX was successfully synthesized and characterized through several physicochemical analyses including MS, FTIR, ^1^H-NMR, ^13^C-NMR, and UV–Vis analysis. Various DFT techniques were employed to supplement the experimental findings of the Schiff base. The overall computational predictions align with the experimental outcomes in a consistent manner.

The characterization of the new Schiff base showed that the selected chemicals had a significant number of phenols. Furthermore, significant antioxidant activity was demonstrated by the DPPH technique, as well as a highly efficient proportion of anti-inflammatory and a moderate anti-hemolytic action. The IC_50_ values indicated that the concentration of chemicals needed to explore these activities were small. It is concluded that chemicals are a good source of a variety of phytochemicals activities. It is concluded that Schiff bases are a good source of a variety of phytochemicals activities.

## Data Availability

The data presented in this study are available on request from the corresponding author.

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
