# Peer review of "Synthesis, Characterization, and DFT Calculations of a New Sulfamethoxazole Schiff Base and Its Metal Complexes"

_materials, 2023, doi:10.3390/ma16145160_

Round 1

Reviewer 1 Report

Al-Hawarin and colleagues prepared a manuscript on synthesis and characterization of new sulfamethoxazole Schiff base and its transition metal complexes. The manuscript contains a large number of punctuation and grammatical errors, which significantly reduce the quality and importance of the work. I recommended that this manuscript can be accepted for publication with major revision. Further efforts are required for improving the quality, grammar and coherence of the manuscript.

1. Such errors include, for example, lines 69, 122, 409 - an extra period; 50 line - 12-14 superscript; 72 - capital letter; 99 - round bracket; 106 - extra zero; 109-111 - words are repeated twice, etc. throughout the text.

2. Lines 392-393 - the sentence should be rewritten and one word of characteristics should be replaced with, for example, properties

3Abstract - it is written that the authors are studying "antibacterial as well as antifungal activity", although the text of the article discusses antioxidant, anti-inflammatory, and anti-hemolytic activities. The first sentence is incomprehensible from the point of view of English. – it should be rewritten “…synthesized from thermal condensation of 3-(2-Furyl) acrolein and sulfamethoxazole (SMX), which is further converted to …”. The conditions under which the metal complexes were obtained are not described. In addition, the authors do not present the results of biological screening, which, in my opinion, should be summarized.

4. Introduction - It is necessary to concretize the research. Why give a definition of anti-inflammatory and other activity, when you need to focus on Schiff bases and what activity they have, as well as cite literature data on the activity of metal complexes, because they are studied in this manuscript.

5. line 110-112 - what does 0H mean? - you need to specify the number of protons.

6. Experimental part - designed differently for compounds - uniformity is needed

7. 11 Literature reference missing from the text - authors should carefully check the list of references

8. Biological activity - it is written very confusingly and incomprehensibly. In the table and in the text, compounds are discussed that have completely different ciphers than those given above in the text, and it is very difficult to understand what they refer to. The authors argue that the amount of phenols plays an important role in the manifestation of antioxidant activity, but the table shows that this is not the case. Therefore, this part needs to be rewritten taking into account these data. It is also not clear why the original Schiff bases were not tested, but only their complexes, which would have been necessary for structure-activity comparisons.

The manuscript contains a large number of punctuation and grammatical errors, which significantly reduce the quality and importance of the work. I recommended that this manuscript can be accepted for publication with major revision. Further efforts are required for improving the quality, grammar and coherence of the manuscript.

Reviewer 2 Report

In this work, a new Schiff base has been synthesized. Characterization techniques including MS, FT-IR, 1H-NMR, 13C-NMR, and UV-Vis were used to analyze the structure of the compound. DFT methods were also used to complement the experimental observation to support the formation of the compound. The overall computational predictions corroborate the experimental results. Solid conclusions were therefore obtained through experimental and theoretical investigations. Here are some comments for this manuscript.

1. Introduction part should be improved to show the significance of this work.

2. Lots of abbreviations are used in this manuscript. However, the authors should present their full name when they first appear. e.g., SMX in Line 45, PVC in Line 48, DFT in Line 75, DMSO in Line 315, PCM in Line 316, et al. Besides, please add the abbreviation TD-DFT for time-dependent DFT in Line 167 since it appears again in Line 332.

3. The Supporting Information can't be found in review system. Please double check whether the Supporting Information has been uploaded or not.

Minor editing of English language is required for this manuscript.

Reviewer 3 Report

PDF file attached.

Round 2

Reviewer 1 Report

The authors took into account all the wishes of the reviewer and the manuscript can be accepted for publication

Author Response

Thank you for your comments